# Trend and Seasonality of Diabetic Foot Amputation in South Korea: A Population-Based Nationwide Study

**DOI:** 10.3390/ijerph19074111

**Published:** 2022-03-30

**Authors:** Hyung-Jin Chung, Dong-Il Chun, Eun Myeong Kang, Keonwoo Kim, Jinyoung Lee, Ye Jin Jeon, Jaeho Cho, Sungho Won, Young Yi

**Affiliations:** 1Department of Orthopedic Surgery, Sanggye Paik Hospital, Inje University College of Medicine, 761-1 Sanggye 7-dong, Nowon-gu, Seoul 01757, Korea; orthoman@paik.ac.kr; 2Department of Orthopaedic Surgery, Soonchunhyang University Seoul Hospital, 59 Daesagwan-ro, Yongsan-gu, Seoul 04401, Korea; orthochun@gmail.com (D.-I.C.); 129741@schmc.ac.kr (E.M.K.); 3Department of Health Administration, Sejong Public Health Center, 32 Daecheop-ro, Jochiwon-eup, Sejong 30029, Korea; jlpt2005@naver.com; 4Department of Statistics, Chung-Ang University, 102 Heukseok-ro, Dongjak-gu, Seoul 06973, Korea; joa.young424@gmail.com; 5RexSoft Corps, Seoul National University Research Park, 1, Gwanak-ro, Gwanak-gu, Seoul 08826, Korea; pinna102@gmail.com (Y.J.J.); sunghow@gmail.com (S.W.); 6Department of Public Health, Yonsei University Graduate School, 50 Yonsei-ro, Seodaemun-gu, Seoul 03722, Korea; 7Department of Orthopaedic Surgery, Chuncheon Sacred Heart Hospital, Hallym University, 77, Sakju-ro, Chuncheon 24253, Korea; hohotoy@nate.com; 8Graduate School of Public Health, Seoul National University, 1, Gwanak-ro, Gwanak-gu, Seoul 08826, Korea; 9Department of Orthopaedic Surgery, Seoul Foot and Ankle Center, Inje University Seoul Paik Hospital, 85, 2-ga, Jeo-dong, Jung-gu, Seoul 04551, Korea

**Keywords:** diabetic foot, diabetic peripheral neuritis, environmental variation, amputation

## Abstract

The number of lower extremity amputations in diabetic foot patients in Korea is increasing annually. In this nationwide population-based retrospective study, we investigated the data of 420,096 diabetes mellitus patients aged ≥18 years using the Korean Health Insurance Review and Assessment Service claim database. We aimed to study the seasonal and monthly trends in diabetic foot amputations in Korea. After applying the inclusion criteria, 8156 amputation cases were included. The analysis showed an increasing trend in monthly amputation cases. In terms of seasonality, the monthly frequency of amputation was commonly observed to be lower in February and September every year. Diabetic foot amputations frequently occurred in March, July, and November. There was no difference between the amputation frequency and mean temperature/humidity. This study is meaningful as it is the first nationwide study in Korea to analyze the seasonal and monthly trends in diabetic foot amputation in relation to climatic factors. In conclusion, we recognize an increased frequency of amputation in March, July, and November and recommend intensive educational program on foot care for all diabetes patients and their caregivers. This could improve wound management and amputation prevention guidelines for diabetic foot patients in the Far East with information on dealing with various seasonal changes.

## 1. Introduction

Diabetes affects people globally and is the leading cause of non-traumatic lower extremity amputations [1,2,3]. Population-based studies have reported large variations in the prevalence and incidence of amputations in people with diabetes, partly related to the different recording periods and inconsistencies in the methods used to define amputation and diabetes [4,5,6]. 

The number of lower limb amputations in diabetic foot patients in Korea is increasing annually [7]. Diabetic foot is accompanied by peripheral neuropathy, peripheral vascular disease, and impaired immunity. All these impairments, in concert, lead to diabetic foot ulceration, sepsis, and eventually, amputation [8].

Some previous research reported that the presentation of patients for diabetic foot amputation was not random; rather, it was related to the weather [9]. There are a limited number of previous studies on the seasonal trend in diabetic foot amputation at the national level, and most of these studies showed an increasing pattern in a certain season due to regional characteristics [10].

When analyzing seasonal characteristics, temperature changes should be considered. Certainly, various other climatic factors in addition to temperature can affect the seasonal characteristics, and these other factors can be understood by considering the geographic characteristics along with the changes in temperature. Comprehending seasonal variations may prevent amputations in the high-risk diabetic foot group.

Health education for diabetic foot disease cannot be overemphasized, and it is necessary to analyze the causes and various factors that aggravate it. Therefore, if you know and apply the difference in seasonal changes in Korea and the Far East, where the four seasons are distinct, you can know and prevent in advance that the patient’s condition may deteriorate depending on the change in temperature or humidity by the season. Leung et al. [11] reported that Hong Kong’s hot and humid weather exacerbated the infection of diabetic foot and promoted amputation. They identified seasonal variations in the frequency of amputations and reported that it was related to warm temperature. In addition, a study in Portugal recommended that intensive care is needed before winter because the amputation rate of diabetic foot disease is high in winter [12].

More severe infections tend to occur in summer. If such a seasonal variation is validated in a certain region, outpatient appointments should be clustered in spring and early summer to alert patients to meticulously check their feet daily and, if needed, seek medical advice earlier. Conversely, if amputations occur frequently in the middle of winter in certain regions, caution should be exercised in early winter [11]. Won et al. [13] reported that amputation risk increases when the daily temperature difference is large in certain geopolitical locations. These factors can be understood by studying the seasonal and geopolitical characteristics.

Meanwhile, in Korea, the frequency of vascular intervention and amputations in diabetic foot patients is continuously increasing, resulting in increased social and economic expenses [14]. Therefore, analyzing the seasonal variation in amputation for diabetic foot patients in Korea could be advantageous in preventing amputations and reducing the socioeconomic burden. Consequently, we aimed to study the seasonal and monthly trends in diabetic foot amputations in Korea. This is the first nationwide study of its kind in Korea.

## 2. Materials and Methods

### 2.1. Study Population and Definition of Diabetic Foot Amputation 

In this nationwide population-based retrospective study, we investigated the data of 420,096 diabetes mellitus patients aged ≥18 years using the Korean Health Insurance Review and Assessment Service (HIRA) claim database. As medical codes that signify diabetic amputation were applied in January 2011, claims data between 2011 and 2018 were included in the current study. During the research period, we included patients in the analysis after a wash-out period of one year. After the HIRA claim database was merged with the diurnal temperature range (DTR) data, 8156 patients who received amputation practice codes after diabetes mellitus diagnosis were included in the main analyses. Definition of diabetes mellitus diagnosis was based on the International Classification of Diseases 10th revision (ICD-10) codes (E10, E11, E12, E13, E14) as the principal diagnosis or additional diagnosis and a prescription of at least one antidiabetic drug including glucose-lowering drugs and insulin in a given year. Antidiabetic drugs were defined using the Anatomical Therapeutic Chemical (ATC) codes (glucose-lowering drug: A10B; insulin: A10A) and medical reviews. 

Diabetic foot amputation related to peripheral artery disease (PAD) was defined as a medical claims code [15]. Therefore, diabetic foot amputation was redefined manipulatively based on the medical practice codes, including percutaneous transluminal angioplasty in other locations (M6597), percutaneous intravascular installation of metallic stents in other locations (M6605), percutaneous intravascular atherectomy (M6620), and amputation of extremities (N0571–N0575). In addition, amputation after the index date of diabetes mellitus was considered as the occurrence of an event, and the earliest amputation was classified as the index event. Finally, 8156 amputation cases were included in this study.

The Institutional Review Board of Seoul Paik Hospital approved this retrospective cohort study (IRB approval no.: PAIK 2020-04-013).

### 2.2. Climate Information during the Research Period 

In this study, the climate information included information on temperature and humidity. For temperature information, the monthly mean daily temperature and DTR were included. DTR was defined as the difference between the daily maximum and minimum temperature ranges. For humidity information, the monthly mean dew point temperature (DPT) and relative humidity (RH), which is the ratio of the humidity ratio of a particular water–⁠air mixture to the saturation humidity ratio at a given temperature, were included. We calculated the monthly mean using a daily measure of the DPT and RH. All climate information was derived from automated surface-observed temperature information in 16 provincial capital cities from the weather data service of the Korea Meteorological Administration (KMA) (data.kma.go.kr). Information on the medical care/treatment institution from the HIRA database and the local information regarding the automated surface observed daily temperature were combined. 

To interpret the time trend or seasonality of the amputation frequency, we calculated the discomfort index (monthly) with temperature and RH as follows:Discomfort Index = 9/5 × temperature − 0.55 (1 − RH) [(9/5 × temperature) − 26] + 32(1)

The study area included 16 provincial capital cities in South Korea between the longitudes 124.60° E to 131.87° E and latitudes 33.11° N to 38.61° N. 

### 2.3. Statistical Analysis

All analyses were conducted using R software (version 3.6.3, R Foundation for Statistical Computing, Vienna, Austria). The monthly average daily climate information was calculated for each variable and further normalized with the annual mean and standard deviation. To assess the time-trend and seasonality of the monthly frequency of diabetic foot amputation, we conducted a time-series decomposition analysis with X11, Seasonal Extraction in ARIMA Time Series (SEAT), and Seasonal and Trend decomposition using Loess (STL) methods (“xts,” “forecast,” and “seasonal” packages were used) [16,17,18,19]. The X11-ARIMA method, which originated in the US Census Bureau and was developed by Statistics Canada, applies integrated autoregressive moving averages processes (ARIMA) to model the original time series and then forecast the value from the fitted ARIMA model to the original series. Finally, the series was enlarged with the forecasted values seasonally adjusted using the standard X11 methods [20]. Similarly, the SEAT method, which was developed by the Bank of Spain [21], was used, as well as the extended version of the standard X11 method. The STL method, which was developed by Cleveland et al. [22], is a robust and widely used method for decomposing the time series. As the name suggests, STL uses the LOESS smoother for estimating nonlinear relationships and decomposing the time series into trend, seasonal, and remainder components.

## 3. Results

### 3.1. Overall and Monthly Trends in Diabetic Foot Amputation

Table 1 and Figure 1 show the monthly trends in diabetic foot amputation and climate information, including temperature and humidity, during the study period (January 2013 to December 2018). The overall trend in diabetic foot amputation cases substantially increased after the medical codes for amputation were assigned. To adjust the linear trend in the amputation cases, we normalized the number of amputation cases with the standard deviation for the year, as shown in Figure 2.

We additionally conducted a time-series decomposition using various decomposition methods: X11, SEAT, and STL (Appendix A) in Figure 3. Regardless of the decomposition method, an increasing trend in monthly amputation cases was observed. Regarding seasonality, the monthly frequency of amputation was commonly observed to be low in February and September every year. Although there was some irregularity in each study year, diabetic foot amputation occurred frequently in March, July, and November.

### 3.2. Comparison between Diabetic Foot Amputation Trend and Climate Information

The monthly frequency of diabetic foot amputation and climate information, including temperature and humidity, during the study period are presented in Figure 2. There was no difference between the amputation frequency and mean temperature/humidity. However, in July, when the seasonality component of the amputation trend was high, both temperature and relative humidity were higher compared with that in other months, and the difference rate of the discomfort index compared with June was also higher (in 2013, mean temperature, 26.03 °C; mean RH, 81.17%; estimated discomfort index, 76.70; difference rate of discomfort index compared with previous month, 9.22%; in 2018, mean temperature, 26.54 °C; mean RH, 78.06%; estimated discomfort index, 77.15; difference rate of discomfort index compared with previous month, 11.13%). In March, the mean temperature and RH were in the normal range, but the DTR and the difference rate of discomfort index were higher compared with that in other months (in 2013, mean DTR, 13.05 °C; difference rate of discomfort index compared with previous month, 20.38%; in 2018, mean DTR, 11.69 °C; difference rate of discomfort index compared with previous month, 24.93). The analysis showed an increasing trend in monthly amputation cases. 

In Figure 3, trends in monthly lower limb amputation rates of diabetic foot patients were analyzed using three different time series (X11-ARIMA, SEATS, STL) decomposition methods. Overall, the three analysis methods showed similar trends. In terms of seasonality, the monthly frequency of amputation was commonly observed to be lower in February and September every year. Diabetic foot amputations frequently occurred in March, July, and November. There was no statistically significant difference between the amputation frequency and mean temperature/humidity.

## 4. Discussion

Management of the various complications of diabetes is vitally important for effective care of diabetic foot. The condition is related to diabetic vasculopathy and neuropathy, as well as various infections that can occur. Diabetic foot care is especially important because of the risk of lower extremity amputation. Patients who have undergone amputation have low survival rates and high socioeconomic costs. From 2011 to 2016, the total number of diabetic foot amputations in Korea increased annually, and the socioeconomic cost increased by 47% [7,14]. This shows a higher rate of increase in Korea compared with that in other countries. Therefore, it is necessary to manage and prevent diabetic foot on a national scale to prevent lower extremity amputations.

Health education for diabetic foot disease is always important regardless of the time, but it is important to analyze and prepare for the contributing factors to amputation in order to cut off the process leading to amputation due to worsening of the disease. As part of this, it is necessary to consider the influence of the season and to manage patients at home, or in healthcare or hospital. Therefore, it is important to analyze and apply seasonal differences.

Appropriate temperature and humidity are key factors in the wound healing process. In particular, changes in temperature and humidity are very important for wound management and hemodynamic stability in diabetic foot patients. Changes in diabetic foot wounds according to the seasonal characteristics of a specific region may occur in various ways. Depending on the seasonal characteristics of the area, various preparations can be made to prevent amputation. However, there are currently few nationwide studies on the seasonal characteristics of diabetic foot amputation.

Gaspar da Rocha et al. reported a seasonal pattern of diabetic foot amputation in Portugal from 2000 to 2015. They reported that the amputation rate in winter was high; therefore, specific measures were needed to prevent diabetic foot disease before winter [12]. Leung et al. [11] in Hong Kong reported that warm temperatures in the area exacerbated infections, leading to amputation. They recommended intensive patient education in late winter and spring to reduce the incidence of amputation.

However, there are no studies in countries with a geological location and distinct seasonal characteristics similar to Korea (middle latitude of the Far East). Portugal, which is adjacent to the European continent and Atlantic Ocean, has a Mediterranean climate, so there is relatively little seasonal variation. Hong Kong’s climate does not drop below 10 °C year-round and remains at approximately 33 °C for the most part. In addition, the relative humidity of the city is maintained at above 70% throughout the year. Leung et al. suggested that more changes could be observed in more dynamic climates. In addition, monthly and time-series analyses were not performed in the previous studies. This study is meaningful in that it is the first nationwide study to analyze the seasonal and monthly trends in diabetic foot amputation in relation to climatic factors.

In this study, the number of amputations increased significantly in March, July, and November. Nevertheless, this study found no significant differences in the frequency, temperature, and humidity of diabetic amputation. However, the highest number of amputations was performed in July, which had the highest temperature and relatively high humidity compared with that in other months. In contrast, March and November had higher daily temperature differences than the other months. The frequency of amputation is high in November and March, before and after the winter when the temperature change is severe, and amputation occurs frequently around midsummer in July, when the temperature is the highest. This tendency is considered to be influenced by season-related climatic factors. Therefore, it can be said that temperature and humidity showed some influences on diabetic foot amputation in terms of season-related climatic factors.

Various studies have been conducted on the effects of temperature on the wound healing process in diabetic foot. Through animal experiments, Won et al. reported that diurnal temperature differences could be the cause of delayed healing of diabetic wounds. They also confirmed the correlation between diurnal temperature range and diabetic foot amputation. Notably, diurnal temperature difference increases at high latitudes in Korea [13].

In this study, few amputations were performed in February and September. We estimated that this could be because the patient’s living environment had stabilized a month since the coldest (January) and the hottest month (August). These months have a more comfortable temperature and humidity than the other months.

Based on these characteristics, we aimed to develop a treatment strategy for the prevention of diabetic foot disease in Korea. Regardless of possible speculation, the existence of seasonal variation remains true. Therefore, it is recommended to provide all diabetics and caregivers in the Far East with an extensive educational program on foot care to minimize diabetic foot ulcers in March, July, and early November. Clinical appointments should also be clustered to facilitate the reinforcement of careful foot care and earlier intervention, if necessary, during the most vulnerable periods. In particular, the indoor environment and the condition of the shoes must be adjusted to provide a consistent environment, regardless of the conditions of the outdoor environment. Efforts should be made to develop other effective tactics to combat nontraumatic amputation for diabetic foot ulcers, a preventable complication. IOT-technology-based temperature and humidity control can be helpful for patients with diabetic foot disease [13].

This study has a few limitations. First, because the medical code of diabetic foot amputation was applied in January 2011, the number of cases was insufficient to estimate the time series trend and seasonality at the beginning of the study. Second, we further calculated the discomfort index to explain the frequent amputation cases with a mean daily temperature of all regions; however, this discomfort index may not be appropriate to represent all 16 regions. Moreover, there were not enough cases of diabetic foot amputation to conduct subgroup analysis. Third, an analysis of the correlation between the seasonal effect on amputation and climate change was limited to temperature, diurnal temperature, humidity, and the discomfort index. Lastly, although we found the seasonality of diabetic foot amputation from the seasonal component using the time series decomposition method, we could not test the significance of seasonality. This is because the seasonal component was estimated using a de-trended series computed by removing the trend component estimated from a specific model in each decomposition method.

This study is the first to analyze the seasonal and monthly changes in lower extremity amputation in patients with diabetic foot in Korea. This could improve wound management and amputation prevention guidelines for diabetic foot patients in the Far East with information on dealing with various seasonal changes. In addition, the treatment can be expected to be more organized in its consideration of seasonal and monthly factors. In the future, based on the national data, individual notifications or home care services for wound stabilization and amputation prevention of diabetic foot patients can be provided. In a follow-up study, the local model will be analyzed by considering the climatic factors of the 16 sub-regions in South Korea.

## 5. Conclusions

According to the national analysis in Korea, the number of amputations in diabetic foot patients showed a steady increase. Although distinct seasonal characteristics could be found in patients with diabetic foot, no clear significance related to climate could be confirmed. However, the monthly analysis results showed significantly more amputations in March, July, and November, and fewer amputations in February and September. The high frequency of amputation in March and November, when the temperature change was severe, and in July, when the temperature was the highest, seems to be influenced by season-related climatic factors. Other geographical locations with an increasing prevalence of diabetic foot amputations should consider a similar nationwide study to decrease the socioeconomic burden on their nation and provide timely guidance and care to diabetic patients.

## Figures and Tables

**Figure 1 ijerph-19-04111-f001:**
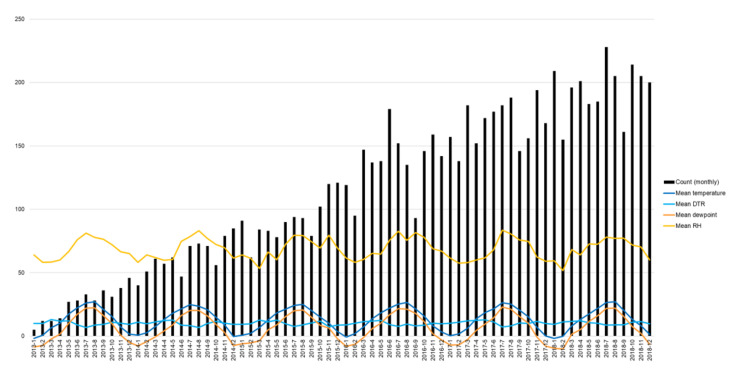
Diabetic foot amputation and climate trend (January 2013−December 2018). (Note) Monthly count of diabetic foot amputation. DTR: diurnal temperature range, difference between the daily maximum and minimum temperature ranges; RH: relative humidity.

**Figure 2 ijerph-19-04111-f002:**
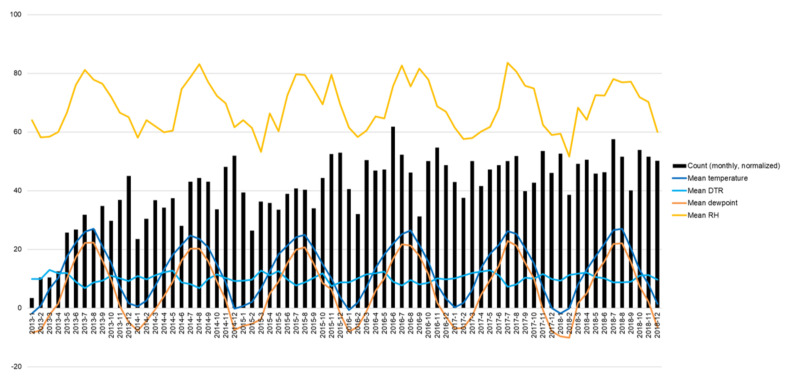
Diabetic foot amputation (normalized) and climate trend (January 2013−December 2018). (Note) Monthly count of diabetic foot amputation normalized with standard deviation for the year. DTR: diurnal temperature range, difference between the daily maximum and minimum temperature ranges; RH: relative humidity.

**Figure 3 ijerph-19-04111-f003:**
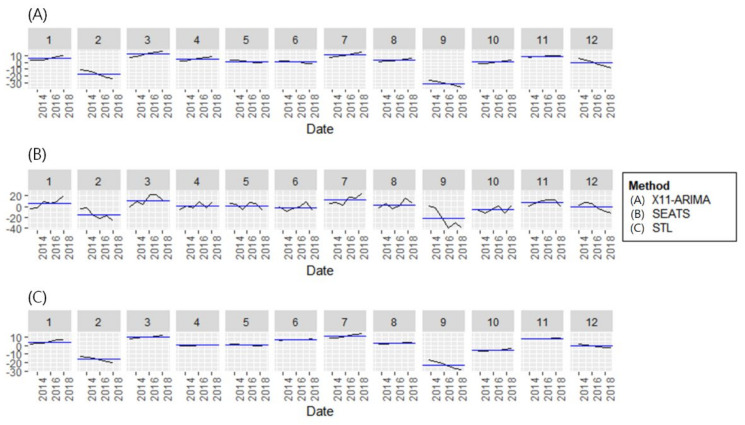
Trends in monthly lower limb amputation rates of diabetic foot patients analyzed using three different time series decomposition methods. (**A**) X11−ARIMA: integrated autoregressive moving averages processes, (**B**) SEATS: seasonal extraction in ARIMA time series, and (**C**) STL: seasonal and trend decomposition using Loess methods. (Note) Black line: Adjusted amputation rate by each methods, Blue line: Mean of adjusted amputation rate by each methods.

**Table 1 ijerph-19-04111-t001:** Monthly diabetic foot amputation frequency and mean temperature and humidity (January 2013–December 2018).

Year-Month	Monthly Frequency	Mean Temperature	Mean DTR	Mean DPT	Mean RH
2013-1	5	−1.79	9.88	−8.43	64.03
2013-2	12	0.81	9.85	−7.56	58.13
2013-3	12	6.58	13.05	−2.29	58.45
2013-4	14	10.19	11.80	1.45	60.04
2013-5	27	17.59	11.96	10.16	66.71
2013-6	28	22.27	8.94	17.27	75.99
2013-7	33	26.03	6.81	22.20	81.17
2013-8	28	27.04	8.77	22.35	77.80
2013-9	36	21.03	9.34	16.17	76.48
2013-10	31	15.41	11.08	9.67	71.91
2013-11	38	7.17	10.17	0.61	66.53
2013-12	46	1.61	9.25	−4.87	65.02
2014-1	40	0.64	10.96	−7.66	58.11
2014-2	51	2.55	9.82	−4.33	64.05
2014-3	61	7.59	11.19	−0.45	62.00
2014-4	57	13.16	12.14	4.09	59.91
2014-5	62	18.12	12.73	8.93	60.42
2014-6	47	21.51	8.75	16.26	74.66
2014-7	71	24.75	8.19	20.29	78.57
2014-8	73	23.59	6.76	20.20	83.07
2014-9	71	20.75	9.84	16.01	76.97
2014-10	56	14.83	11.54	9.09	72.30
2014-11	79	8.91	10.25	2.86	69.73
2014-12	85	−0.25	9.25	−7.47	61.56
2015-1	91	0.72	9.37	−6.05	64.06
2015-2	62	2.12	9.67	−5.37	61.39
2015-3	84	6.72	12.72	−3.79	53.27
2015-4	83	12.56	11.16	5.01	66.35
2015-5	78	18.33	12.69	8.95	60.28
2015-6	90	21.33	9.77	15.29	72.37
2015-7	94	24.11	7.69	19.94	79.66
2015-8	93	24.99	8.80	20.71	79.41
2015-9	79	20.27	10.41	14.93	74.55
2015-10	102	14.94	11.84	8.51	69.42
2015-11	120	10.18	7.48	6.37	79.53
2015-12	121	3.71	8.73	−2.00	69.46
2016-1	119	−0.69	8.80	−8.04	61.48
2016-2	95	1.87	10.07	−6.40	58.26
2016-3	147	7.17	11.51	−1.19	60.45
2016-4	137	13.66	11.85	5.86	65.30
2016-5	138	18.33	12.47	10.16	64.58
2016-6	179	22.02	9.01	16.83	75.63
2016-7	152	25.21	7.66	21.69	82.70
2016-8	135	26.45	9.52	21.22	75.53
2016-9	93	21.48	8.02	17.81	81.66
2016-10	146	15.80	8.68	11.46	77.84
2016-11	159	7.97	10.18	1.85	68.77
2016-12	142	3.25	9.82	−3.02	66.91
2017-1	157	0.27	10.13	−7.05	61.54
2017-2	138	1.74	11.04	−6.74	57.58
2017-3	182	6.22	11.95	−2.75	57.93
2017-4	152	13.67	12.45	4.46	60.14
2017-5	172	18.46	12.83	9.60	61.79
2017-6	177	21.52	11.16	14.33	68.03
2017-7	182	26.24	7.30	22.88	83.61
2017-8	188	25.32	8.08	21.28	80.60
2017-9	146	20.47	10.40	15.40	75.75
2017-10	156	15.26	9.99	10.18	74.83
2017-11	194	6.91	11.62	−0.73	62.38
2017-12	168	0.01	9.86	−7.95	59.01
2018-1	209	−1.87	9.29	−9.63	59.40
2018-2	155	−0.08	11.22	−10.12	51.67
2018-3	196	8.03	11.69	1.58	68.30
2018-4	201	13.14	12.11	5.05	64.13
2018-5	183	17.59	10.62	11.67	72.54
2018-6	185	21.92	10.09	15.96	72.44
2018-7	228	26.54	8.71	21.90	78.06
2018-8	205	27.04	8.79	22.10	76.95
2018-9	161	20.33	8.93	15.68	77.19
2018-10	214	13.05	11.08	7.35	71.79
2018-11	205	8.17	11.31	2.20	70.25
2018-12	200	1.40	9.64	−6.50	60.00

(Note) The monthly mean value of each climatic parameter was calculated. DTR: diurnal temperature range, difference between the daily maximum and minimum temperature ranges; DPT: dew point temperature; RH: relative humidity.

## Data Availability

Data may be obtained from a third party and are not publicly available.

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
