# Peer review of "Trend and Seasonality of Diabetic Foot Amputation in South Korea: A Population-Based Nationwide Study"

_ijerph, 2022, doi:10.3390/ijerph19074111_

Round 1

Reviewer 1 Report

The authors presented impressive research in their manuscript „Trend and Seasonality of Diabetic Foot Amputation in South Korea: a population-based nationwide study. “ They showed that foot amputations are constantly increasing in South Korea. The authors did not find an association between foot amputations in diabetic patients with seasonal characteristics. They found more amputations in March, July, and November and fewer in February and September. The cause is the effect of monthly climate change.

Figure 3 It is not clear, so that it could be omitted.

Author Response

* The authors did not find an association between foot amputations in diabetic patients with seasonal characteristics.
> The association of climatic factors with seasons was further explained.
"Considering that the frequency of amputation is high in November and March, when the temperature change is severe before and after of the winter, and amputation occurs a lot around midsummer in July, when the temperature is the highest. This tendency is considered to be influenced by season-related climatic factors."
* Figure 3 It is not clear, so that it could be omitted.
> a time-series decomposition methods was needed. so It was mentioned in 3.1 and 3.2.

Reviewer 2 Report

This article analyzes 8,156 cases of 420,096 diabetic patients in Korea, and finds that amputation cases are likely to occur in February, July and November. Congratulations to the author for this important discovery, but the health education for diabetes is uninterrupted throughout the year, not because which season is more important and which season is less important. Therefore it is not recommended for publication.

Author Response

  • Please see the attachment.
  • the health education for diabetes is uninterrupted throughout the year, not because which season is more important and which season is less important.
    > Health education for diabetic foot disease is always important regardless of the time, so we mantioned it and explained the need to consider the influence of the seasons.

Reviewer 3 Report

The authors carried out a brief report in order to study the seasonal and monthly trends in diabetic foot amputations in Korea.

The article is interesting, and it addresses a current topic, however, there are a few areas that need to be added/improved before publishing: it requires a further literature review to establish state-of-the-art with respect to the area of investigation, in particular on the seasonality of amputations that occur in other countries, also considering the reversed seasons in the various areas of the world. Bibliography should be extended

Author Response

  • it requires a further literature review to establish state-of-the-art with respect to the area of investigation, in particular on the seasonality of amputations that occur in other countries, also considering the reversed seasons in the various areas of the world. Bibliography should be extended
  • > Additional explanation is provided in the introduction. However, there have been limitations as not much research has been done on amputation and seasonality in other various countries.

Round 2

Reviewer 2 Report

The previous review opinion held that health education is something that needs to be carried out continuously, not which months need to be paid attention to which months are not used. However, after being revised by the comments of the other two reviewers, the logic of the article is stronger and the conclusion is reasonable. Therefore it is recommended to publish.